

# Measurement report: Ammonia in Paris derived from ground-based open-path and satellite observations

Camille Viatte[1], Nadir Guendouz[1], Clarisse Dufaux[1], Arjan Hensen[2], Daan Swart[3], Martin Van Damme[4,5], Lieven Clarisse[4], Pierre Coheur[4], and Cathy Clerbaux[1,4]

[1]LATMOS/IPSL, Sorbonne Université, UVSQ, CNRS, Paris, France;

[2]Netherlands Organisation for Applied Scientific Research (TNO), P.O. Box 15, 1755 ZG, Petten, the Netherlands

[3]National Institute for Public Health and the Environment (RIVM), Bilthoven, the Netherlands;

[4]Université libre de Bruxelles (ULB), Spectroscopy, Quantum Chemistry and Atmospheric Remote Sensing (SQUARES), Brussels, Belgium;

[5]BIRA-IASB - Belgian Institute for Space Aeronomy, Brussels, Belgium;

*Correspondence:* Camille Viatte (camille.viatte@latmos.ipsl.fr)



**Abstract**
Ammonia ($NH_3$) is an important air pollutant which, as precursor of fine particulate matter, raises
public health issues. This study analyzes 2.5-years of $NH_3$ observations derived from ground-based
(miniDOAS) and satellite (IASI) remote sensing instruments to quantify, for the first time, temporal
variabilities (from interannual to diurnal) of $NH_3$ concentrations in Paris.
The IASI and miniDOAS datasets are found to be in relatively good agreement (R>0.70) when
atmospheric $NH_3$ concentrations are high and driven by regional agricultural activities. Over the
investigated period (January 2020 – June 2022), $NH_3$ average concentrations in Paris measured by the
miniDOAS and IASI are 2.23 µg.m$^{-3}$ and 7.10x10$^{15}$ molecules.cm$^{-2}$, respectively, which are lower or
equivalent to those documented in urban areas. The seasonal and monthly variabilities of $NH_3$
concentrations in Paris are driven by sporadic agricultural emissions influenced by meteorological
conditions, with $NH_3$ concentrations in spring up to 2 times higher than in other seasons.
The potential source contribution function (PSCF) reveals that the close (100-200km) east and
northeast regions of Paris constitute the most important potential emission source areas of $NH_3$ in the
megacity.
Weekly cycles of $NH_3$ derived from satellite and ground-based observations show different ammonia
sources in Paris. In spring, agriculture has a major influence on ammonia concentrations and, in the
other seasons, multi-platform observations suggest that ammonia is also controlled by traffic-related
emissions.
In Paris, the diurnal cycle of $NH_3$ concentrations is very similar to the one of $NO_2$, with morning
enhancements coincident with intensified road traffic. $NH_3$ evening enhancements synchronous with
rush hours are also monitored in winter and fall. $NH_3$ concentrations measured during the weekends
are consistently lower than $NH_3$ concentrations measured during weekdays in summer and fall. This is
a further evidence of a significant traffic source of $NH_3$ in Paris.



## 1. Introduction

Ammonia (NH₃) is an air pollutant which is involved in important environmental and health issues [Rockström et al., 2009]. It is a highly reactive gas, with a lifetime of a few hours to a few days [Evangeliou et al., 2021; Dammers et al., 2019], capable of reacting with nitrogen oxides (NOₓ) and sulfur oxides (SOₓ) to form fine particulate matter composed of ammonium nitrate and ammonium sulfate [Sutton et al., 2013]. The formation of fine particles plays a major role in the degradation of air quality, as they are the cause of respiratory and cardiovascular diseases [Pope III et al., 2009].

Models have difficulty predicting events of particulate pollution associated with NH₃ since ground-based atmospheric observations of this gas are still relatively sparse [Nair and Yu, 2020] and difficult to implement [Twigg et al., 2022; von Bobrutzki et al., 2010]. To our knowledge, only six countries in the world (United States, China, the Netherlands, United Kingdom, Belgium, and Canada) have dedicated NH₃ observations in their atmospheric monitoring networks. This poses a problem for long-term monitoring of pollution and the implementation of emission reduction policies.

Global population growth causes increased food demand leading to higher ammonia emissions from intensive agricultural production systems [Fowler et al., 2013]. Global NH₃ emissions have increased by more than 80% between 1970 and 2017 [McDuffie et al., 2020]. In Europe, a substantial increase in nitrate and ammonium concentrations in the composition of fine articles has been observed for several years in the early spring when fertilizer applications intensify [Favez et al., 2021]. In addition, the share of emissions related to road traffic is also increasing because of popularization of catalytic converters in car engines [Zhang et al., 2021]. In France, 98% of ammonia comes from agricultural activities, via decomposition and volatilization of nitrogen fertilizers (34%) and animal waste (64%), the rest are from industry, road traffic and residential heating [CITEPA, 2022]. In the Ile-de-France region (Paris greater area), the share of agriculture is lower (75%) due to a higher contribution of traffic and residential sectors (13% and 12%, respectively [AirParif, 2022]). NH₃ emissions from road traffic are very poorly quantified and may be a larger than expected source in urban areas [Pu et al., 2023; Chatain et al., 2022; Cao et al., 2021; Roe et al., 2004; Sutton et al., 2000].

Monitoring NH₃ is therefore essential, especially in urban areas such as in Paris, where particulate pollution episodes are monitored almost every spring [Viatte et al., 2021; Viatte et al., 2020; Petetin et al., 2016].

Global scale measurement of atmospheric ammonia is possible via soundings from several satellite-borne instruments such as AIRS [Warner et al., 2016], CrIS [Shephard and Cady-Pereira, 2015], and IASI [Clarisse et al., 2009]. Satellite measurements of atmospheric ammonia allow a description of its spatial distribution with global coverage. The detection of the multi-year evolution of concentrations is possible, as well as the detection of emission sources at the kilometer scale [Van Damme et al., 2018], and even the quantification of their variabilities [Van Damme et al., 2021; Dammers et al., 2019]. Remote sensing data are also used as a mean to estimate ammonia emission inventories [Marais et al., 2021; Cao et al., 2020; Fortems-Cheiney et al., 2020].

Quantifying and analyzing temporal NH₃ variabilities at different scales (diurnal, weekly, seasonal, and interannual) helps to improve emission inventories and air quality forecasts [Cao et al., 2021]. Diurnal NH₃ variability, which is rarely measured, is particularly crucial because atmospheric models have difficulty representing it [Lonsdale et al., 2017]. NH₃ concentrations increase during the day due to the temperature dependence of emissions, but there may be many other factors at play influencing the



diurnal variability of NH₃ concentrations in the atmosphere, such as transport, boundary layer height,
deposition, fertilizer application time, road traffic emissions, and the interaction of all these factors
[Sudesh and Kulshrestha, 2021; Osada, 2020; Wang et al., 2015]. The diurnal variability of NH₃, which
is still largely missing from the ground and satellite observations, provides valuable information
regarding sources, surface exchange, deposition, gas-particle conversion, and transport of NH₃
[Clarisse et al., 2021].
In this work, we present 2.5-years of atmospheric NH₃ concentrations measured in Paris using the
synergy of ground-based and IASI satellite observations to quantify NH₃ variabilities at different time
scales.

## 2. Methodology

### 2.1. mini-DOAS

The miniDOAS (Diffential Optical Absorption Spectroscopy) is a state-of-art instrument suitable for NH₃
monitoring [Berkhout et al., 2017] since it performs accurate high temporal resolution measurements
(every hour, day and night) [Volten et al., 2012]. It has been designed and developed by the National
Institute for Public Health and the Environment (RIVM, Netherlands) to be part of the Dutch National
Air Quality Monitoring Network [Berkhout et al., 2017]. The miniDOAS is an active remote sensing
instrument based on open-path differential absorption spectrometry. It uses a xenon lamp which emits
a UV light, ammonia having a strong absorption band in the UV between 200 and 230 nm. The UV light
beam travels along an optical path of 20 m, at the end of which there is a reflector which reflects the
UV light and sends it back to the spectrometer/receiver. The Beer-Lambert law is used to quantify the
extinction at the absorption wavelengths of ammonia to retrieve atmospheric ammonia
concentrations [Volten et al., 2012]. The miniDOAS can measure a wide range of ammonia
concentrations (from 0.5 to 200 μg.m⁻³) day and night with no sampling artifacts, since it is not using
any filter or inlet unlike other instruments [Caville et al., 2023; von Bobrutzki et al., 2010]. Estimated
errors are $4.10^{-3}$ μg.m⁻³ on hourly measurements [Volten et al., 2012]. Using ammonia measurements
performed from the miniDOAS at the QUALAIR super-site (40 meters above ground level,
https://qualair.fr/index.php/en/english/) in the Paris city-center, the NH₃ contribution in particulate
pollution events that occurred during the 2020 COVID lockdown has been demonstrated [Viatte et al.,
2021].

### 2.2. IASI

The Infrared Atmospheric Sounding Interferometer (IASI, [Clerbaux et al., 2009]) was launched first in
2006 as part of the Metop satellite series to monitor atmospheric composition twice a day (at 9:30 and
21:30) globally. IASI measures atmospheric spectra in the thermal infrared region with an elliptical
pixel footprint of 12 × 12 km at nadir and 20 × 39 km at the far end of the swath. In this study, we use
NH₃ columns derived from IASI morning (9:30) overpasses onboard Metop B and C from January 2020
to June 2022. When comparing IASI and miniDOAS NH₃ concentrations in Paris, we have selected
coincident observations made within the same hour. In this work, we use version 3 of the ANNI-NH3
reanalyzed dataset [Van Damme et al., 2021; Guo et al., 2021; Viatte et al., 2022].

### 2.3. Meteorological data from ERA-5

Meteorological parameters originate from the ERA-5 database of the European Centre for Medium-
Range Weather Forecasts (ECMWF, [Hersbach et al., 2020]). It is constituted from observations



recalibrated on global data assimilation models at a 30km spatial resolution. In this work, we used the
hourly data of the temperature at 2 m, the precipitation, the u and v components of the wind at 100
m and the height of the boundary layer, taken from the grid cells in which Paris is located.

### 2.4. Back-trajectories and Potential Source Contribution Factor (PSCF) analysis

To study the transport affecting concentration of ammonia in Paris, we use the Hybrid Single-Particle
Lagrangian Integrated Trajectory model (HYSPLIT, [Stein et al., 2015]) to calculate backward
trajectories of air masses ending at altitudes of 100 m (above sea level which corresponds to the
altitude of the miniDOAS location) between January 2020 and June 2022.
Meteorological data used in the runs are from the National Centers for Environmental Prediction
(NCEP) / National Center for Atmospheric Research (NCAR) reanalysis at 2.5-degree global latitude-
longitude projection. We ensure by visual inspections that the back trajectories using a 2.5° resolution
meteorological dataset are similar to using a finer meteorological dataset at 0.25° resolution (GFS).
Due to the short and highly variable lifetime of $NH_3$, ranging between 2-4 hours [Dammers et al., 2019]
and 12-hours [Evangeliou et al., 2021], we simulated an average 6-h backward trajectories with an
interval of one hour. Combining the hourly $NH_3$ observations from the miniDOAS, the potential
emission sources of $NH_3$ were analyzed. The Potential Source Contribution Factor (PSCF) method
[Malm et al., 1986] is used to identify source regions affecting air quality in term of $NH_3$ concentration
in Paris between January 2020 and June 2022. This method is now commonly used in atmospheric
science [Wang et al., 2023; Qadri et al., 2022; Martino et al., 2022; Biuki et al., 2022; Ren et al., 2021;
Zachary et al., 2018; Jeong et al., 2011] and combines the concentration dataset with air parcel back-
trajectory to identify preferred pathways producing high observed $NH_3$ concentrations in Paris. The
larger PSCF (range: 0–1), the greater contribution of the pollution region to the atmospheric pollutants
at the receptor site.



**3. Results**

**3.1. Comparison of NH₃ concentrations between IASI and mini-DOAS**

The 2.5-years mean NH$_3$ total column distribution around Paris derived from IASI from January 1$^{st}$ 2020 to May 31$^{st}$ 2022 is shown in Figure 1 (top panel). To obtain averages at a high resolution needed for city-scale studies, we used the oversampling method that takes into account the real elliptical sizes of each IASI pixel [Van Damme et al., 2018]. Hot spots of ammonia are found around Paris in agricultural areas, especially in the Champagne-Ardennes region between Troyes and Reims cities [Viatte et al., 2020].

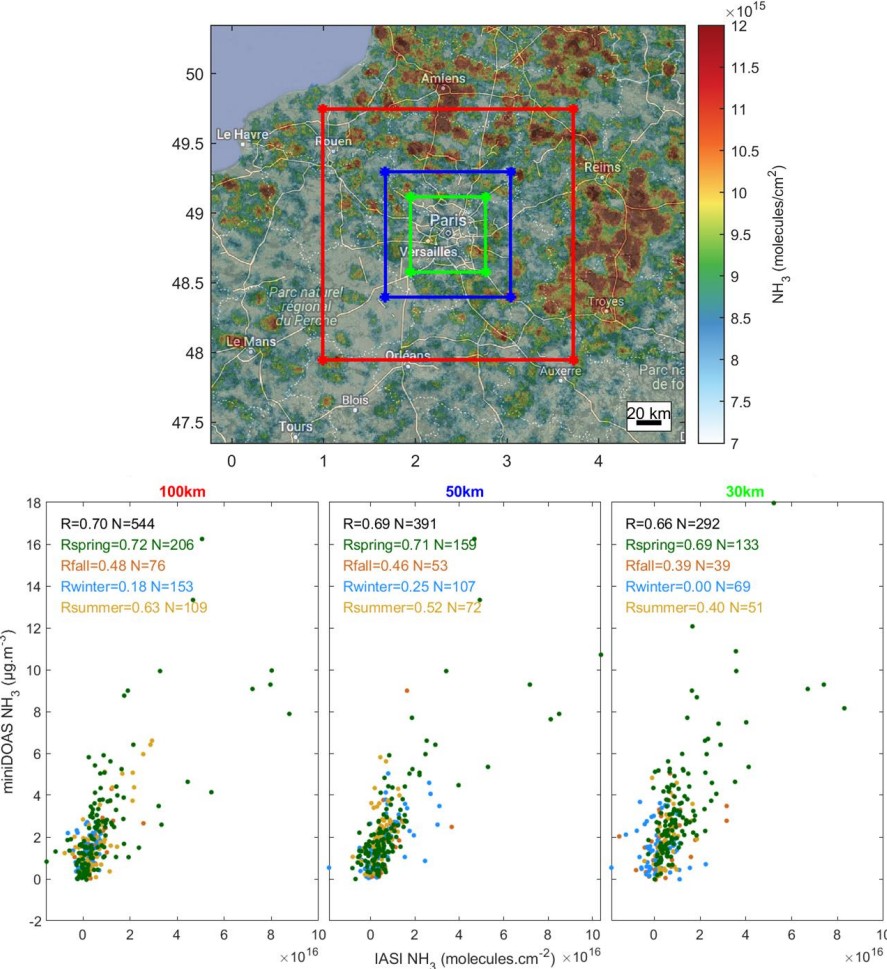

*Figure 1: Top panel: 2.5-years average of IASI NH₃ column distributions (from January 1$^{st}$ 2020 to May 31$^{st}$ 2022). Bottom panel: miniDOAS ground-based NH₃ concentrations (µg.m⁻³) versus IASI-retrieved NH₃ column concentrations (molecules.cm⁻²) per season for different spatial criteria from Paris city center where the miniDOAS is located (100km in red box, 50km in blue box, and 30km green box).*



NH₃ has a short atmospheric lifetime which is why we only compare miniDOAS data recorded within
the same hour as the IASI morning overpass time.  The IASI-retrieved column (in molecules.cm$^{-2}$) and
the miniDOAS ground-based concentrations (µg.m$^{-3}$) are qualitatively compared to assess the spatial
criteria (100km in red box, 50km in blue box, and 30km green box) and the season for which both
datasets are in best agreement. In this study we are not converting IASI columns to surface
observations since it introduces additional errors and does not change the correlation as explained in
[Van Damme et al., 2015].
Overall, the miniDOAS and IASI NH₃ concentrations are in moderate agreement with Pearson
correlation [Akoglu et al., 2018] of 0.70, 0.69, and 0.66 when considering IASI pixels within a 100km,
50km, and 30km box around Paris, respectively. The number of pairs is, however, reduced by a factor
of two when considering IASI pixels in a 100km versus a 30km box around Paris. All correlations are
significant (p-value < 0.05) except in winter for the 100km and 30km boxes, and in fall for the 30km
box. The best agreement between the miniDOAS and IASI is in spring, with Pearson correlations ranging
from 0.72 to 0.69 (green points in scatter plots of Figure 1). This period corresponds to high
atmospheric NH₃ concentrations when spreading practices occur in the surrounding agricultural
regions of Paris [Viatte et al., 2022]. In fall and summer, the Pearson correlation coefficients range
from 0.63 to 0.40 between IASI and the miniDOAS for all boxes sizes. In winter, the agreements are
poor between the miniDOAS and IASI because NH₃ concentrations are weak and IASI is less sensitive
to lower atmospheric layers when thermal contrast is low [Van Damme et al., 2014]. In addition, we
demonstrate that correlations between satellite and ground-based NH₃ observations are independent
of atmospheric temperature and planetary boundary layer height (PBLH, Figure S1).
A trade-off between good correlations and keeping a sufficient number of collocations is found when
comparing NH₃ concentrations from ground-based measurements located in the Paris city-center with
the IASI dataset in a 50 km box. We chose for the rest of the analysis IASI dataset within the 50 km box
to analyze spatiotemporal variabilities of NH₃ in Paris.
**3.2. Impact of agriculture on NH₃ concentrations in Paris**

180            **3.2.1 2.5-years of NH₃ measurements in Paris**

Here, we investigate temporal variabilities of NH₃ using 2.5-years of hourly measurements from
January 1$^{st}$ 2020 to May 31$^{st}$ 2022 (Figure 2). The miniDOAS was working almost full time during this
period with 16 888 hourly measurements, out of the 21 145 possible. The missing data is due either to
some technical issues during warm conditions (malfunctioning aircondition in August 2021) or due to
its removal from the QUALAIR facility for field measurement campaigns (from September 15$^{th}$ 2021 to
November 24$^{th}$ 2021).  Over the 16 888 hourly NH₃ measurements, average errors are 2.8 10$^{-3}$ µg.m$^{-3}$
with maximum values occurring when signal is low due to a transient poor alignment (such as in April
2020, yellow dots in Figure 2).



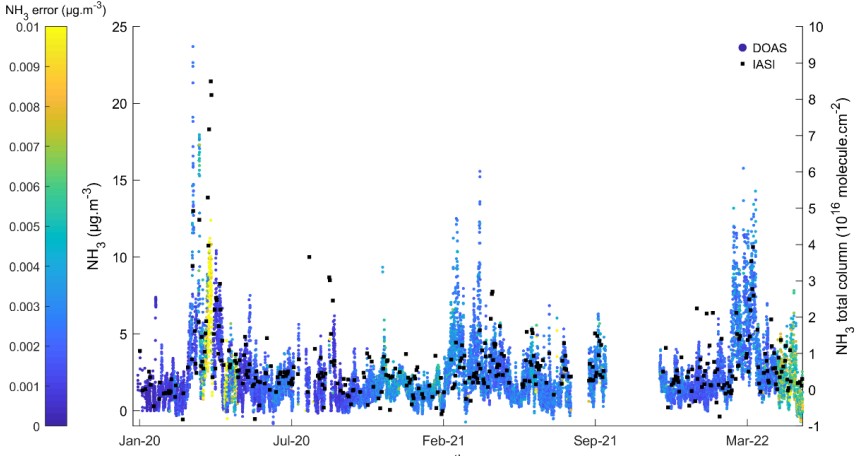


*Figure 2: Timeseries of hourly NH₃ concentrations (in μg.m⁻³) color coded by the errors on measurements derived from the miniDOAS located in Paris, and IASI NH₃ total columns (in black, molecule.cm⁻²) observed in a 50km box centered in Paris from January 1ˢᵗ 2020 to May 31ˢᵗ 2022.*

The measurements made by the miniDOAS over the period January 2020 - June 2022 (N=16 888) show an average ammonia concentration of 2.23 μg.m⁻³ in Paris over this period, with a standard deviation of 2.02 μg.m⁻³, indicating a high NH₃ variability. In comparison, the average concentration measured by the miniDOAS in an agricultural site at Grignon [Loubet et al., 2022] in September-October 2021 (France) is 6.52 ± 8.44 μg.m⁻³ [Claville et al., 2023], almost three times higher than in Paris. The relatively low concentrations observed in Paris are explained by the distance to the major emission sources which are related to agricultural activities. Ammonia concentrations measured in Paris are on average lower or equivalent to those documented in urban areas such as Beijing (China, 21 ± 14 ppb corresponding to 14.7 ± 10 μg.m⁻³ from January 2018 to January 2019, [Lan et al., 2021]), Shanghai (China, 6.2 ± 4.6 ppb which corresponds to 4.3 ± 3.2 μg.m⁻³ from July 2013 to September 2014, [Wang et al., 2015]), Rome (Italy, 1.2–21.6 μg.m⁻³ between May 2001 and March 2002, [Perrino et al., 2002]), Milan (Italy, 4.4–13.4 μg.m⁻³ between 2007 and 2019, [Lonati et al., 2020]), Louisville (Unites-States, 2.2–5.2 μg.m⁻³ from June to August 2011, [Li et al., 2017]) and Toronto (Canada, 2.5 ppb which corresponds to 1.75 μg.m⁻³ from 2003 to 2011, [Hu et al. 2014]).

The miniDOAS and IASI coincident measurements show relatively low interannual variability (Table 1). NH₃ annual concentrations measured by the miniDOAS are 2.06 ± 2.09 μg.m⁻³ and 2.04 ± 1.56 μg.m⁻³ for 2020 and 2021, respectively. The higher mean and standard deviation in 2022 (2.91 ± 2.40 μg.m⁻³ for the miniDOAS) compared to the other years can be due the fact that measurements are performed from January to June only. IASI NH₃ total columns around Paris exhibit a higher NH₃ annual concentration and standard deviation in 2020 compared to the other years because of high pollution events occurring in spring during the 2020-COVID lockdown [Viatte et al., 2021].



*Table 1: Average NH₃ concentration, standard deviation, and number of observations for 2020, 2021*
*and part of 2022 derived from coincident measurements of the miniDOAS and IASI (50 km box around*
*Paris).*

| years | 2020 | | 2021 | | 2022 | |
|---|---|---|---|---|---|---|
| | miniDOAS | IASI (50km) | miniDOAS | IASI (50km) | miniDOAS | IASI (50km) |
| NH₃ concentration (μg.m⁻³ or molecules.cm⁻²) | 2.06 | $8.60 \times 10^{15}$ | 2.04 | $5.48 \times 10^{15}$ | 2.91 | $6.76 \times 10^{15}$ |
| Standard deviation (μg.m⁻³ or molecules.cm⁻²) | 2.09 | $1.58 \times 10^{16}$ | 1.56 | $5.69 \times 10^{15}$ | 2.40 | $9.35 \times 10^{15}$ |
| Number of observations | 7164 | 166 | 6182 | 134 | 3542 | 91 |


### 3.2.2 Seasonal and monthly NH₃ variabilities in Paris

Unlike the weak interannual variability of NH₃ concentrations in Paris, both ground-based (miniDOAS)
and satellite (IASI) measurements reveal high seasonal variabilities of NH₃ concentrations (Figure 3). In
spring, NH₃ concentration measured in Paris by the miniDOAS and IASI are on average
$3.34 \pm 2.67$ μg.m⁻³ and $1.21 \times 10^{16} \pm 1.57 \times 10^{16}$ molecules.cm⁻², respectively. These springtime NH₃
concentrations are enhanced by a factor of two compared to the other seasons, which is consistent
with the fertilizer application periods over the nearby agricultural fields. Both datasets show that NH₃
concentrations in March and April are 2 to 3 times higher than the other months. Precipitation for
these months is also lower than in February on average (see supplementary Figure S2).

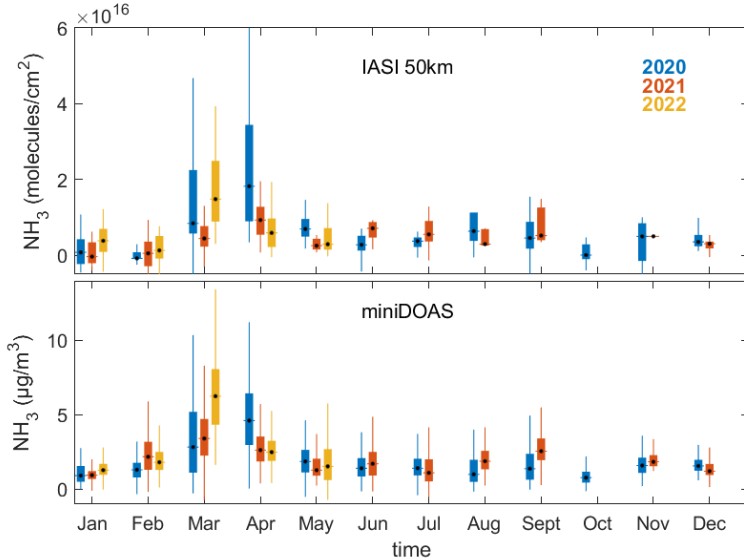


*Figure 3: Monthly NH₃ concentrations color coded by the year of measurements (2020 in blue, 2021 in*
*orange, and 2022 in yellow) derived from IASI (top panel, in molecules.cm⁻²) in a 50km box around Paris*
*and the ground-based miniDOAS instrument (bottom panel, in μg.m⁻³) located in Paris city-center. One*
*note that IASI observations are only considered when a miniDOAS observation is available within the*
*same hour than IASI overpass.*



When considering each year of measurement separately, we notice that the timing of the maximum
NH$_3$ concentrations is variable. In 2020, the maximum is reached in April with averaged NH$_3$
concentrations of 4.76 ± 2.48 µg.m$^{-3}$ (miniDOAS) and 2.90x10$^{16}$ ± 2.85x10$^{16}$ molecules.cm$^{-2}$ (IASI),
whereas in 2022 the maximum appears in March with a monthly NH$_3$ concentration of 6.42 ± 2.46
µg.m$^{-3}$ and 1.72x10$^{16}$ ± 1.04x10$^{16}$ molecules.cm$^{-2}$ derived from the miniDOAS and IASI, respectively.
Meteorological conditions influence the timing of the agricultural practices (farmers do not spread
their fertilizer when it rains), NH$_3$ volatilization from the soil to the atmosphere (higher temperature
favors NH$_3$ volatilization [Sutton et al., 2013]), and the transport of NH$_3$ over Paris.
In April 2020, NH$_3$ concentrations observed by IASI and the miniDOAS are high compared to April 2022.
In April 2020, precipitation is low (0.3 mm compared to 0.75mm in April 2022) and the monthly
averaged atmospheric temperature is on 3 to 5°C higher than in 2021 and 2022 (Figure S2). This could
explain why NH$_3$ concentrations are higher in April 2020 than in 2022. Similarly, the lower ammonia
concentration recorded in March 2021 compared to March 2022 is likely explained by higher
precipitation (0.09 mm) and a lower temperature (of 2°C on monthly average) than in March 2022.
In 2021, a second NH$_3$ enhancement is measured in September by the miniDOAS (2.73 ± 1.14 µg.m$^{-3}$)
and IASI (7.93x10$^{15}$ ± 4.64x10$^{15}$ molecules.cm$^{-2}$). The pronounced seasonal variability can be explained
in the first order by the practices of the farmers. In most European countries, strict regulations are
applied in term of the timing of fertilizer application [Ge et al., 2020]. In France, it is forbidden to spread
nitrogen fertilizers in winter months (between November 30th and February 15th, [Ludemann et al.,
2022]) depending on fertilizer and land/crop types.
Overall, the seasonal and monthly variabilities of NH$_3$ concentrations in Paris are dominated by
agricultural activities and meteorological conditions.

### 3.2.3 Potential Source Contribution Function (PSCF) analysis for NH$_3$ concentrations

To determine the origin of the NH$_3$ measured in Paris, the Potential Source Contribution Function
(PSCF) is used. The PSCF analysis, as well as the IASI NH$_3$ maps, are shown for the investigated period
(January 2020 – June 2022, Figure 4 upper panels), and for springs 2020, 2021, and 2022 (Figure 4,
three lower panels).
Over the whole timeseries, the northeast (100 km from Paris in the Aisne department of France) and
east (70km from Paris in the "Seine et Marne" department) locations are found to affect the NH$_3$
concentrations observed in the city between January 2020 and June 2022. These areas are indeed
source regions of NH$_3$ according to coincident IASI observations (Figure 4, upper panels). According to
wind fields parameters derived from ERA-5 over Paris (not shown here), the winds from the south are
more intense (up to 18 m.s$^{-1}$) and are related to lower ammonia concentrations (between 0 and 4 µg.m$^{-3}$
$^{-3}$). The northern winds are on average weaker (maximum around 12 m.s$^{-1}$) and are associated with
higher ammonia concentrations. In particular, for the northeast section the measured NH$_3$
concentration is found to exceed 8 µg.m$^{-3}$.
According to the PSCF analysis, the main sources of NH$_3$ from agricultural activities are found in the
close areas of Paris (within 100 and 200 km from Paris city-center) mainly from the east and northeast
directions. In France, the averaged utilized agricultural area per department in 2020 is 64.5 ha (Agreste
– Recensements agricoles, https://stats.agriculture.gouv.fr/cartostat/#c=home). The highlighted



departments by the PSCF analysis are ranked to have the most cultivated areas in France with 141.5
ha for Seine et Marne, 124.4 ha for Oise, and 110.4 ha for Aisne departments for instance.

*Figure 4: Potential Source Contribution Function (PSCF, left) and IASI NH₃ total columns (right, in*
*molecules.cm⁻²) The top raw is the January 2020 to June 2022 average, and the 3 lower panels are for*
*springs 2020, 2021, and 2022. The blue dot indicates the location of Paris.*
In spring, when NH₃ concentrations are significantly higher in Paris (Figure 3) and in the surroundings
(Figure 4 three lower right panels), the PSCF analysis show that the northeast and southeast regions
are the major sources of the observed NH₃ concentrations in Paris. In spring 2020, NH₃ columns are
higher than in spring 2021 and 2022, according to IASI observations. The main sources of NH₃ in spring
2020 are pronounced in the nearby east-northeast areas (at 50 km from Paris in the surrounding
departments of Seine et Marne, Oise, and Val d'Oise). In spring 2021, IASI observations reveal lower
NH₃ columns than in 2020 and 2022 and the sources of NH₃ concentrations in Paris are in the
surrounding regions of Paris (100 km in all directions). In spring 2022, the northeast pathway is
highlighted similarly to spring 2020 but with a contribution of the southeast region as well.



### 3.3 Effect of road traffic on NH₃ variability in Paris

#### 3.3.1 Weekly cycle of NH₃ concentrations

The weekly cycles of ammonia concentrations measured in Paris by the miniDOAS and IASI over the studied timeseries are presented in Figure 5 (black bars, top panels). Both datasets show an increase of ammonia concentrations during the week, reaching a maximum on Thursday (2.21 µg.m$^3$ for the miniDOAS and 5.90x10$^{15}$ molecules.cm$^{-2}$ for IASI).

The weekly cycle of IASI measurements in Paris is almost analogous to the one observed over European agricultural areas with low concentrations observed on Mondays and an accumulation of ammonia during the week [Van Damme et al., 2022]. In addition, the IASI NH₃ weekly cycle averaged over 2.5-years of measurements in Paris is very similar to the NH₃ weekly cycle measured in spring (Figure 5) when agricultural activities intensify. Monitoring similar NH₃ weekly variability in the urban area of Paris demonstrates that agricultural activities in the surrounding areas control the variability of ammonia in Paris on average over the whole season.

The NH₃ weekly cycle observed over 2.5-years of measurements from the ground-based miniDOAS and the IASI satellite observations show, however, relatively low NH₃ concentrations on Saturday and Sunday. The cycle is less pronounced for IASI measurements. Ammonia concentrations observed over the weekend by the miniDOAS and IASI are lower by 25% and 20% compared to NH₃ concentrations averaged over the weekdays in Paris.

When considering intraweek variabilities by seasons (Figure 5, four lower panels), one can observe that both IASI and the miniDOAS dataset reveal similar NH₃ weekly cycles. The NH₃ miniDOAS measurements and coincident IASI total columns measured in a 50km box around Paris exhibit lower concentrations over the weekends compared to weekdays for all seasons, except in spring for which higher NH₃ concentrations are found on Wednesday and Sunday. In spring, the miniDOAS and IASI measure a difference of NH₃ concentrations averaged over the weekends compared to weekdays of +1% and -7%, respectively. In fall, summer, and winter, the miniDOAS (IASI) instrument measure a decrease of NH₃ concentrations between weekends and weekdays of 70% (34%), 42% (28%), and 27% (53%) respectively.

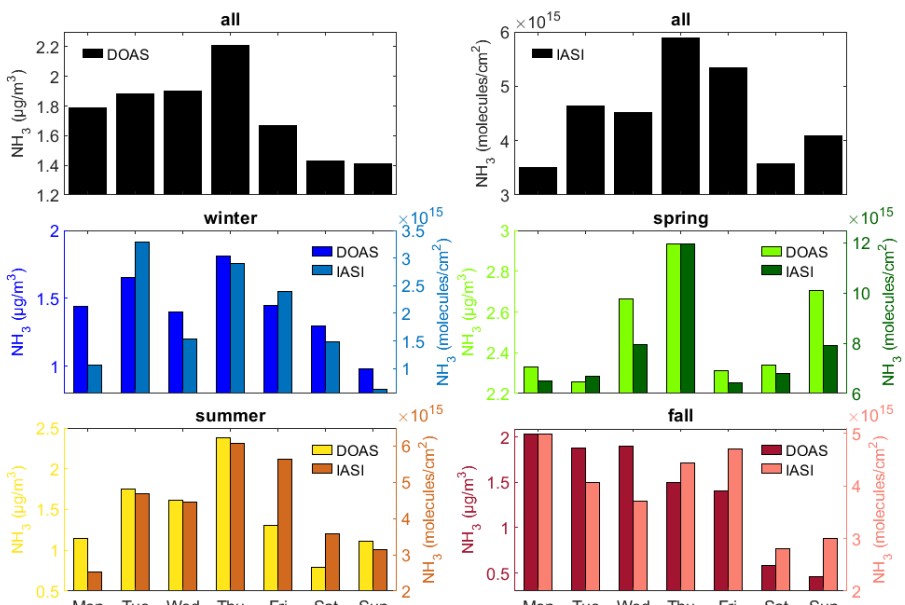

316

*Figure 5: Day of the week NH₃ concentrations derived from the miniDOAS (µg.m⁻³) and IASI*
*(molecules.cm⁻²) in Paris for the investigated period (January 2020 to May 2022, top panels), and for*
*different seasons (winter in blue, spring in green, summer in brown and yellow, and fall in red and pink*
*bars).*

Comparing these weekly variabilities with those of the weekly flow of cars in Paris (Figure S3), the same
pattern is clearly highlighted with a stable number of cars per hour from Monday to Friday (around
640) and a decrease of 14% over the weekends.
We can make the hypothesis that during all seasons except spring, the influence of the agricultural
practices on the variability of ammonia in Paris is less pronounced, revealing NH₃ contribution from
the traffic source. Since the road traffic intensity is constant throughout the year in Paris, the
proportion of ammonia emitted from road traffic is proportionally higher outside the fertilization
period.

### 3.3.2 Diurnal cycle of NH₃ concentration in Paris

With the high temporal resolution of the mini-DOAS acquisitions, the diurnal variability of NH₃
concentration is assessed in Paris using, for the first time, a quasi-continuous (temporal coverage of
80%) and a relatively long timeseries of 2.5-years of NH₃ observations.
Hourly NH₃ concentrations measured by the miniDOAS from January 2020 to May 2022 are shown in
Figure S4. It shows a marked diurnal variability of NH₃, with a decrease of about 30% in the middle of
the day (around 14:00 LT) compared to the night, then an increase in the afternoon to reach again a
maximum during the night.



Note that this diurnal variability of NH₃ measured by the miniDOAS is different than the one reported
during springtime pollution episodes from a ground-based Fourrier Transform InfraRed spectrometer
located in the suburbs of Paris [Kutzner et al., 2021]. While measured integrated NH₃ total columns
show an intraday increase until late afternoon, the miniDOAS measures NH₃ concentrations varying in
opposition to the boundary layer height (Figure S4). This reflects the dynamical effect of the boundary
layer height, which is controlled by atmospheric temperature, on the dilution of pollutants
concentrations measured close to the surface. Such effect is also seen with surface measurements of
NO₂ concentrations in Paris (Figure S4).

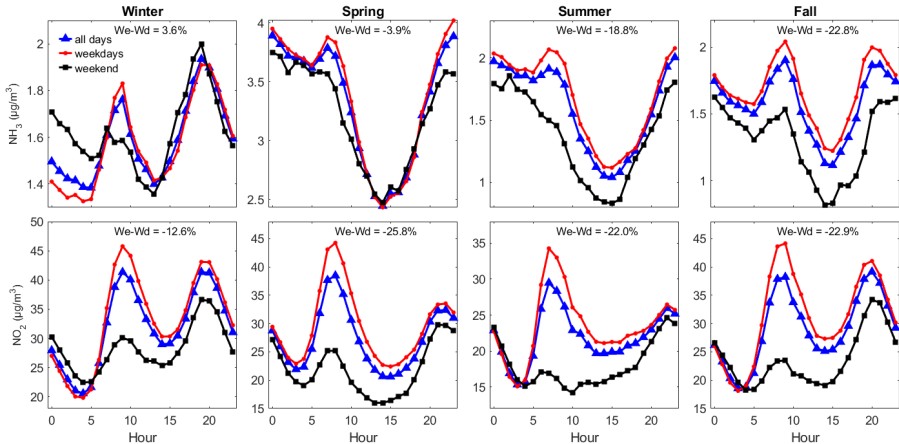


*Figure 6: Diurnal variability of NH₃ (upper panels) and NO₂ (lower panels) concentrations measured by*
*the miniDOAS and Airparif in (µg.m⁻³) averaged by seasons using 2.5-years of measurements in Paris.*
*Hours are indicated in local time. The diurnal variability of NH₃ and NO₂ are shown in blue lines when*
*considering all days, in red lines for weekdays, and in black lines for weekends.*
The diurnal variability of NH₃ concentrations presents an increase in the morning visible for all seasons
(Figure 6). Between 5:00 and 8:00, road-traffic in Paris increases by a factor 4 (Figure S3) and NH₃
concentrations rise by more than 20% in winter and fall, and about 3% in summer and spring.
To verify the hypothesis that road traffic is responsible for these morning enhancements, NO₂ diurnal
variability is also shown in Figure 6 (lower panels). In Paris, NO₂ is considered as a proxy for road traffic
emissions [Pazmino et al., 2022]. For all seasons, morning enhancements of NO₂ concentrations related
to intensified road traffic emissions are coincident with morning enhancements of NH₃ concentrations.
Similarly, enhancements of NO₂ and NH₃ concentrations are observed during the evenings (20:00 to
22:00 LT) in winter and fall only. In spring, agriculture which is the overall dominant source of ammonia
in Paris, prevents from monitoring NH₃ emitted from road traffic. Conversely, in fall and winter, the
relative share of agriculture is weaker, and the peaks of NH₃ concentrations during rush hours (morning
and evening) are clearly observed by the miniDOAS.
Diurnal variability of NH₃ and NO₂ concentrations averaged during weekdays (red lines) and weekends
(black lines) are shown in Figure 6. NO₂ concentrations are systematically lower during weekends by
12.6%, 25.8%, 22.0%, and 22.9% in winter, spring, summer, and fall respectively, compared to



weekdays. Similarly, diurnal cycle of NH₃ concentrations averaged during weekends are constantly
lower than NH₃ concentrations averaged during weekdays in summer and fall by 22.0% and 22.9%.
This highlights the importance of traffic emissions of NH₃ in such urban area of Paris, detected by
ground-based measurements when agricultural practices are reduced in the surrounding region.
These results are consistent with previous studies showing the importance of NH₃ emissions from
traffic in urban areas, such as in Rome (Italy, [Perrino et al., 2002]), in Beijing (China, [Ianniello et al.,
2010]), in Shanghai (China, [Wang et al., 2015]), and in Manchester (United Kingdom, [Whitehead et
al., 2004]) for instance. These emissions have gradually become another major contribution of
ammonia pollution in urban areas in the United States and China [Sun et al., 2017]. Ammonia emissions
from road vehicles are shown to be underestimated in the United Kingdom [Farren et al., 2020] and in
densely-populated areas in China [Wen at al., 2022]. In France, NH₃ levels measured at a traffic site are
significantly higher than those observed in a background site [Chatain et al., 2022]. Our results in Paris
confirm that traffic has a significant contribution to atmospheric nitrogen budgets and stress the need
for further NH₃ monitoring in urban sites.

## 4. Conclusion

Atmospheric variabilities of NH₃ concentrations in Paris are assessed using joined observations of
ground-based (miniDOAS) and satellite (IASI) remote sensing observations from January 2020 to June
2022. We present the first relatively long (2.5-years) and continuous record of hourly NH₃
concentrations in Paris to determine temporal variabilities of ammonia at different scales (from
interannual to diurnal variability) to unravel emission sources (traffic and agriculture).
Qualitative comparison of NH₃ derived from the ground-based miniDOAS located in Paris city-center
and IASI satellite observations reveals an overall moderate agreement with Pearson's correlation
coefficients of 0.66, 0.69 and 0.70 when considering IASI observations in a 100km, 50km, and 30km
box around Paris. The best agreement between both datasets is found during springtime when NH₃
concentrations are 2 to 3 times higher than during the other seasons due to spreading practices
occurring in the surrounding agricultural regions of Paris. Overall, agricultural activities driven by
favorable meteorological conditions (high temperature and low precipitation) control the seasonal and
monthly variabilities of NH₃ in Paris. The PSCF analyses indicate that the close east and northeast
agricultural regions (within 100 and 200 km from Paris city-center) affect the most the NH₃ budget in
Paris.
Road-traffic emissions are noticeable in the weekly NH₃ cycles measured by satellite and ground-based
instruments, when agricultural related emissions are weak. Ammonia concentrations observed over
the weekend by the miniDOAS and IASI are lower by 25% and 20% compared to NH₃ concentrations
averaged over the weekdays. In addition, diurnal cycles of NH₃ concentrations in Paris are similar to
NO₂ and reveal coincident enhancements during rush hours. Further long-term NH₃ monitoring in
urban areas is needed to better estimate NH₃ emissions from the on-road sector and their impact on
secondary particle formation.
We have shown that the planetary boundary layer height greatly influences diurnal variabilities derived
from surface measurements. Future work will be carried to compare these NH₃ datasets in Paris to
atmospheric model outputs to evaluate the timing and the absolute value of emission inventories, as



well as the partition between NH₃ emission sectors (traffic vs. agriculture). The launch of the
geostationary MTG satellite carrying the hyperspectral sounder IRS, scheduled for 2024, will offer
unprecedent atmospheric observations with a spatial resolution of 4 km × 4 km (at the Equator) and a
high temporal resolution (every 30 minutes over Europe). These new observations will improve our
understanding of the diurnal variability of ammonia, and it will be a great addition to the miniDOAS
and IASI observations.

### Data availability

The IASI NH₃ dataset used in this study are available via the Zenodo repository
https://doi.org/10.5281/zenodo.7962362 (Viatte, 2023). The miniDOAS data are available here
https://iasi-ft.eu/products/nh3_minidoas/ (Viatte, 2023). The ERA-5 data are available via the Climate
Data              Record              (CDR)              Copernicus              website
https://cds.climate.copernicus.eu/cdsapp#!/search?text=ERA5%20back%20extension&type=dataset
(C3S CDS, 2023). The potential source contribution function is available via the Meteothink.org
http://meteothink.org/docs/trajstat/pscf.html (Wang et al., 2009). Last access to all URLs: 23 May
423  2023.


### Author contributions

CV and NG designed the project. MVD and LC provided the IASI data. AH, AW, DS helped with the
miniDOAS installation and data acquisition. CV and CD analyzed the data. CV and CD wrote the
manuscript draft. All the co-authors reviewed and edited the manuscript. CC wrote proposals to
financially support the miniDOAS.

### Competing interests

The authors declare that they have no conflict of interest.

### Acknowledgments

IASI is a joint mission of EUMETSAT and the Centre National d'Etudes Spatiales (CNES, France). The IASI
Level 1C data are distributed in near real time by EUMETSAT through the EUMETCast system
distribution. The authors acknowledge the AERIS data infrastructure (https://www.aeris-data.fr) for
providing access to the IASI Level 1 radiance and Level 2 NH₃ concentration data used in this study.
CNES and the AC-SAF (CDOP3) project provided financial support for the miniDOAS acquisition. We
thank the NOAA's Air Resources Laboratory (ARL) for providing the HYSPLIT model. Research at ULB
was supported by the Belgian State Federal Office for Scientific, Technical and Cultural Affairs (Prodex
HIRS) and the Air Liquide Foundation (TAPIR project). LC is Research Associate supported by the Belgian
F.R.S.-FNRS. MVD is supported by the FED-tWIN project ARENBERG funded via the Belgian Science
Policy Office (BELSPO).



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
