# Peer review of "Measurement report: Ammonia in Paris derived from ground-based open-path and satellite observations"

_EGUsphere, 2023_

## Referee Comment (RC1)

**Review of "Measurement report: Ammonia in Paris derived from ground-based open-path and satellite observations"**

This paper analyzes NH₃ measurements from the IASI instrument on the Metop-B and Metop-C platforms and from a miniDOAS instrument located in the center of Paris. The two datasets are shown to be correlated overall and also on seasonal and weekly time scales. These analyses suggest that traffic is the very significant source of NH₃ in Paris, except in the spring. A source contribution approach (PSCF) uses the miniDOAS data and ERA-5 meteorological fields to demonstrate that transport from agricultural regions to the northeast, east and southeast dominates in the spring. The contribution from traffic is further confirmed by comparisons with weekly and daily traffic and NO₂ data.

This is a clearly organized and well written paper that adds to the growing body of work on NH₃ sources and variability in large urban areas. It requires only minor revisions to be acceptable for publication.

**Technical comments**

1. The authors need to make clear in the section 2.1 that there is only one miniDOAS instrument and provide its location. This becomes apparent later in the paper, but it would make the earlier sections clearer if stated up front.
2. On line 62 comment on why there are pollution episodes every spring in Paris.
3. Is the center of the IASI pixel used to determine if it falls within the averaging box? If yes, please state this.
4. While the authors do discuss the trade off between analysis box size and number of IASI samples, to justify their choice of a 50 km window, given the high spatial variability of NH3 it is certain that the air masses sampled by IASI and the miniDOAS are very different. Could the high correlations in spring between the two datasets be attributed to the fact that the main sources of NH3 are outside Paris? Do the authors think that the lower correlations in spring and fall could be in part due to urban sources playing a larger role, which may drive greater variability within the 50 km box? Any further comments on the spatial variability issue?
5. On line 173 the authors state that correlations between the IASI and miniDOAS observations are independent of atmospheric temperature and PBL height, and point to Figure S1. This figure is not very convincing. And it contradicts the statement in section 3.3.2 (and in the conclusions) that the miniDOAS observations are sensitive to the PBL height, as they measure a local concentration , while the FTIR (and IASI) are not, as they measure columns. Thus it is difficult to believe that the correlation does not depend on PBL height. I suggest either removing the statement, or clarifying it by calculating and showing the correlations within temperature and PBL bins.
6. The PSCF is shown for all data points and for spring. I suggest showing this function for summer and fall also. If it shows that the NH3 sources in these seasons are within the urban area, it would provide an additional argument to the importance of urban sources.

**Minor revisions**

Line 1 : Title is odd: I would remove the "measurement report" phrase.

Line 36: …which plays a role in …

Line 62: … are observed almost every spring …

Line 69: … their variability …

Line 99: … day and night, and does not suffer from sampling artifacts, since it does not use a filter or inlet, unlike other commonly used instruments (*also provide examples of instruments, not just references)*

Line 101: Using ammonia measurements obtained from the miniDOAS at the QUALAIR super-site (40 meters above ground level, https://qualair.fr/index.php/en/english/) in the Paris city-center, Viatte et al. (2021) demonstrated the contribution of NH3 to particulate pollution events that occurred ….

Line 117:  … 2020], which is built from observations recalibrated into a global assimilation model at a 30 km resolution.

Line 132: Using the hourly …

Line 145: we used the oversampling method described by van Damme et al. (2018), which takes into account the real elliptical sizes of each IASI pixel…
   *Are the values in this map ever used in the analysis? Please clarify.*

Line 212: … because of the multiple high pollution events …

Line 230: city center. Note that …

Line 258: …, in the spring of 2020, 2021 …

Line 278: … top row …

Line 279: *Make blue dot larger*

Line 317: *suggestion for Figure 5: always use the darker color for miniDOAS*

Line 380: …assessed using joint observations …

Line 390: …due to fertilizer spreading …

Line 393: The PSCF analyses indicate that the agricultural regions to the east and northeast within 100 to 200 km from Paris city-center have the greatest impact on the NH3 budget in Paris.

Figure S3: *cite source for these data.*

---

## Author Comment (AC1)

**Review of "Measurement report: Ammonia in Paris derived from ground-based open-path and satellite observations"**

Referee: This paper analyzes $NH_3$ measurements from the IASI instrument on the Metop-B and Metop-C platforms and from a miniDOAS instrument located in the center of Paris. The two datasets are shown to be correlated overall and also on seasonal and weekly time scales. These analyses suggest that traffic is the very significant source of NH3 in Paris, except in the spring. A source contribution approach (PSCF) uses the miniDOAS data and ERA-5 meteorological fields to demonstrate that transport from agricultural regions to the northeast, east and southeast dominates in the spring. The contribution from traffic is further confirmed by comparisons with weekly and daily traffic and $NO_2$ data.

This is a clearly organized and well written paper that adds to the growing body of work on $NH_3$ sources and variability in large urban areas. It requires only minor revisions to be acceptable for publication.

Authors: we would like to thank the referee for his insightful and constructive comments. We have addressed the referees 'suggestions and questions to improve the manuscript. The original review comments are shown in black, our responses are shown in blue.

**Technical comments**

1. The authors need to make clear in the section 2.1 that there is only one miniDOAS instrument and provide its location. This becomes apparent later in the paper, but it would make the earlier sections clearer if stated up front.

We have clarified this by adding these following sentences at the beginning of section 2.1 : "$NH_3$ concentrations are measured since January 2020 in the Paris city-center (48.8°N, 2.3°E) using the ground-based miniDOAS instrument located at the QUALAIR super-site (40 meters above ground level, https://qualair.fr/index.php/en/english/). This dataset constitutes the only current continuous (day and night) $NH_3$ observations at high temporal frequency representative of the Paris megacity."

2. On line 62 comment on why there are pollution episodes every spring in Paris.

Several publications have shown that particulate matter pollution episodes in springtime are found to be almost annually frequent in Paris and often associated with emissions from agricultural activities in the areas surrounding the agglomerations. We have listed other significant papers and added this statement to the revised manuscript: "Monitoring $NH_3$ is therefore essential, especially in urban areas such as in Paris, where particulate pollution episodes are monitored almost every spring [Viatte et al., 2022] and often associated with emissions from agricultural activities in the surrounding areas [Viatte et al., 2021; Kutzner et al., 2021; Viatte et al., 2020; Petetin et al., 2016; Petit et al., 2015]."

3. Is the center of the IASI pixel used to determine if it falls within the averaging box? If yes, please state this.

Indeed, the center of IASI pixels determines if it falls within the averaging box. We have clarified this in section 2.2 of the revised manuscript: "we have selected coincident observations made within the same hour and the center of the IASI pixels was used to determine the distance between the miniDOAS and IASI measurements".

4. While the authors do discuss the trade-off between analysis box size and number of IASI samples, to justify their choice of a 50 km window, given the high spatial variability of NH$_3$ it is certain that the air masses sampled by IASI and the miniDOAS are very different. Could the high correlations in spring between the two datasets be attributed to the fact that the main sources of NH3 are outside Paris? Do the authors think that the lower correlations in spring and fall could be in part due to urban sources playing a larger role, which may drive greater variability within the 50 km box? Any further comments on the spatial variability issue?

We have added the following statements in the revised manuscript to discuss the spatial variability of NH$_3$ observed with the two datasets:

"The high correlation in spring between the two datasets can be attributed to two factors: 1) NH$_3$ concentrations are higher and therefore the signal measured by the two instruments are larger leading to a better correlation from the wide range of NH$_3$ concentrations (0-18 µg.m$^{-3}$ for the miniDOAS and 0-1.10$^{16}$ molecules.cm$^{-2}$ for IASI, Figure 1) and 2) the high amount of NH$_3$ emitted in spring in the surrounding regions due to fertilizer applications can be transported to Paris [Viatte et al., 2022; Viatte et al., 2021] resulting in high correlations between the ~12-km IASI footprints and the local miniDOAS observations."

"In fall and summer, the lower correlations between the ground-based and the satellite NH$_3$ observations could reveal specific NH$_3$ sources in the close vicinity of the miniDOAS which might be not representative of the IASI pixels size."

5. On line 173 the authors state that correlations between the IASI and miniDOAS observations are independent of atmospheric temperature and PBL height, and point to Figure S1. This figure is not very convincing. And it contradicts the statement in section 3.3.2 (and in the conclusions) that the miniDOAS observations are sensitive to the PBL height, as they measure a local concentration, while the FTIR (and IASI) are not, as they measure columns. Thus it is difficult to believe that the correlation does not depend on PBL height. I suggest either removing the statement, or clarifying it by calculating and showing the correlations within temperature and PBL bins.

We agree that figure S1 is not convincing, thus we have removed this statement and this figure in the revised supplementary information and manuscript documents.

6. The PSCF is shown for all data points and for spring. I suggest showing this function for summer and fall also. If it shows that the NH$_3$ sources in these seasons are within the urban area, it would provide an additional argument to the importance of urban sources.

The goal of the PSCF analysis is to attribute the main geographical sources of NH$_3$ emissions that affect the urban air quality of Paris. As mentioned previously, it is known that agricultural practices are enhanced in spring over our region of study. This is why we have performed PSCF analysis in spring for three different years (2020, 2021, and 2022) to study the interannual variability of the major NH$_3$ source regions that affect the NH$_3$ budget in Paris.

As the referee's suggested, we have also performed the PSCF analysis per seasons, as shown in Figure R1 of this document. We clearly see that the PSCF shows an extended area of the regional contribution in spring compared to the other seasons. In addition, NH$_3$ concentrations and variability in Paris in spring are at least two times higher than for the other seasons, proving the importance of the surrounding agriculture practices in the high NH$_3$ budget in Paris. However, even if we see that PSCF are significantly reduced around Paris during fall, winter, and summer compared to spring, we cannot conclude from this figure that the urban area is the unique contribution of NH$_3$ budget in Paris. One

reason might be that we have used 6-hours back-trajectories to compute the PSCF analysis for the whole dataset. This value was chosen based on the large range of NH$_3$ lifetimes found in the literature, which are from 2 hours to few days [Hertel, et al., 2012; Behera et al., 2013; Haugustaine et al., 2014; Whitburn et al., 2016; Van Damme et al., 2018; Dammers et al., 2019; Evangeliou et al., 2021; Luo et al., 2022]. Since NH$_3$ lifetime depends on atmospheric chemical loss and wet deposition (which are linked to atmospheric temperature and humidity), it will thus be different depending on the season. Choosing a constant value of NH$_3$ lifetime throughout the year will bias the length of the back-trajectories depending on the seasons.

[Figure]

Figure R1: Potential Source Contribution Function (PSCF) per season. The blue dot indicates the location of Paris.

Hertel, O.; Skjøth, C.; Reis, S.; Bleeker, A.; Harrison, R.; Cape, J.N.; Fowler, D.; Skiba, U.; Simpson, D.; Jickells, T.Governing processes for reactive nitrogen compounds in the European atmosphere. Biogeosciences 2012, 9, 4921 4954, doi:10.5194/bg-9- 4921-2012.

Behera, S.N.; Sharma, M.; Aneja, V.P.; Balasubramanian, R. Ammonia in the atmosphere: a review on emission sources, atmospheric chemistry and deposition on terrestrial bodies. Environ. Sci. Pollut. Res. 2013, 20, 8092-8131, doi:10.1007/s11356-013- 2051-9.

Hauglustaine, D.A.; Balkanski, Y.; Schulz, M. A global model simulation of present and future nitrate aerosols and their direct radiative forcing of climate. Atmos. Chem. Phys. 2014, 14, 11031-11063, doi:10.5194/acp-14-11031-2014.

Whitburn, S.; Van Damme, M.; Clarisse, L.; Turquety, S.; Clerbaux, C.; Coheur, P.F. Doubling of annual ammonia emissions from the peat fires in Indonesia during the 2015 El Niño. Geophys. Res. Lett. 2016, 43, 11,007-011,014, doi:10.1002/2016gl070620.

Van Damme, M.; Clarisse, L.; Whitburn, S.; Hadji-Lazaro, J.; Hurtmans, D.; Clerbaux, C.; Coheur, P.-F. Industrial and agricultural ammonia point sources exposed. Nature 2018, 564, 99-103, doi:10.1038/s41586-018-0747-1.

Dammers, E.; McLinden, C.A.; Griffin, D.; Shephard, M.W.; Van der Graaf, S.; Lutsch, E.; Schaap, M.; Gainairu-Matz, Y.; Fioletov, V.; Van Damme, M., et al. NH3 emissions from large point sources derived from CrIS and IASI satellite observations. Atmos. Chem. Phys. 2019, 19, 12261-12293, doi:10.5194/acp-19-12261-2019.

Evangeliou, N.; Balkanski, Y.; Eckhardt, S.; Cozic, A.; Van Damme, M.; Coheur, P.-F.; Clarisse, L.; Shephard, M.W.; Cady-Pereira, K.E.; Hauglustaine, D. 10-year satellite-constrained fluxes of ammonia improve performance of chemistry transport models. Atmos. Chem. Phys. 2021, 21, 4431-4451, doi:10.5194/acp-21-4431-2021.

Luo, Z.; Zhang, Y.; Chen, W.; Van Damme, M.; Coheur, P.-F.; Clarisse, L. Estimating global ammonia (NH3) emissions based on IASI observations from 2008 to 2018. Atmos. Chem. Phys. 2022, 22, 10375-10388, doi:10.5194/acp-22-10375-2022.

**Minor revisions**

The minor revisions have all been considered in the revised manuscript.

Line 1: Title is odd: I would remove the "measurement report" phrase.

We agree but it is mandatory to publish in the ACP measurement report journal: "The title must clearly reflect the manuscript type and start with "Measurement report:" (https://www.atmospheric-chemistry-and-physics.net/about/manuscript_types.html)"

Line 36: …which plays a role in …

Line 62: … are observed almost every spring …

Line 69: … their variability …

Line 99: … day and night, and does not suffer from sampling artifacts, since it does not use a filter or inlet, unlike other commonly used instruments (also provide examples of instruments, not just references)

Line 101: Using ammonia measurements obtained from the miniDOAS at the QUALAIR supersite (40 meters above ground level, https://qualair.fr/index.php/en/english/) in the Paris citycenter, Viatte et al. (2021) demonstrated the contribution of $NH_3$ to particulate pollution events that occurred ….

Line 117: … 2020], which is built from observations recalibrated into a global assimilation model at a 30 km resolution.

Line 132: Using the hourly …

Line 145: we used the oversampling method described by van Damme et al. (2018), which takes into account the real elliptical sizes of each IASI pixel… Are the values in this map ever used in the analysis? Please clarify.

To clarify, we have added this sentence to the revised manuscript: "All IASI maps shown in this study were computed using this methodology".

Line 212: … because of the multiple high pollution events …

Line 230: city center. Note that …

Line 258: …, in the spring of 2020, 2021 …

Line 278: … top row …

Line 279: Make blue dot larger

Line 317: suggestion for Figure 5: always use the darker color for miniDOAS

Line 380: …assessed using joint observations …

Line 390: …due to fertilizer spreading …

Line 393: The PSCF analyses indicate that the agricultural regions to the east and northeast within 100 to 200 km from Paris city-center have the greatest impact on the $NH_3$ budget in Paris.

Figure S3: cite source for these data.

---

## Author Comment (AC2)

**Review of "Measurement report: Ammonia in Paris derived from ground-based open-path and satellite observations"**

Referee: The manuscript discusses relatively long-term $NH_3$ measurements from the IASI instrument and a miniDOAS instrument located in the center of Paris. Correlation, seasonal and weekly cycles are discussed as well as the potential sources of the $NH_3$ emissions.

The manuscript is well-written and well structured. The findings do not introduce substantial new discoveries. The finding that traffic is a significant source of $NH_3$ in the Paris region would be more convincing with a better placement of the miniDOAS or when $NH_3$ measurements from more stations would have been included. Such measurements seem to be available in the Paris region. Although I find the study a bit lightweight for ACP, I still recommend its publication. However, revisions (detailed below) need to be conducted in the paper before publication. Please note that all questions/comments below should not only be addressed in the author's reply, but also in the manuscript.

Authors: we would like to thank the referee for his insightful and constructive comments. We have addressed the referees 'suggestions and questions to improve the manuscript. The original review comments are shown in black, our responses are shown in blue.

**General comments**

p.4 l. 103: One of the main comments is that a clear motivation is needed why you put the instrument at an altitude of 40 m above ground level. You are far less sensitive to the traffic related emissions you want to monitor as well? In the introduction (p.3 l.59) it is stated that this is badly monitored.

Our main goal in this work was not to only study $NH_3$ traffic related emissions in Paris. Our first motivation is to assess the temporal evolution of $NH_3$ concentrations over the megacity using the first relatively long-term $NH_3$ observations (2.5 years, day and night) and disentangle the different sources when studying $NH_3$ budget at different timescales. This is why we chose to install the mini-DOAS instrument in the Paris city-center (48.8°N, 2.3°E) at the QUALAIR super-site at 40 meters above ground level (https://qualair.fr/index.php/en/english/) so that the $NH_3$ observations footprint is representative of the Paris capital.

We have clarified this by adding this following sentence at the beginning of section 2.1 : "$NH_3$ concentrations are measured since January 2020 in the Paris city-center (48.8°N, 2.3°E) using the ground-based miniDOAS instrument located at the QUALAIR super-site (40 meters above ground level, https://qualair.fr/index.php/en/english/). This dataset constitutes the only current continuous (day and night) $NH_3$ observations at high temporal frequency representative of the Paris megacity."

Fig.1 top panel: It is difficult to see the $NH_3$ distribution in the Paris area on this map. It would be helpful to also provide a zoom for the red, blue and green box (maybe in appendix).

As mention in the manuscript, we demonstrated that that agricultural activities in the surrounding areas control the variability of ammonia in Paris on average over the whole season. This map is computed using the whole IASI dataset (2.5-years) which reveals the main $NH_3$ agricultural sources

around Paris. We have tried to improve this figure (see Figure R1) but we decided to keep the old one in the revised document.

[Figure]

Figure R1: 2.5-years average of IASI NH₃ column distributions (from January 1st 2020 to May 31st 2022).

Fig.1 lower panel: I'm not sure if applying these different spatial criteria make much sense. Also it would be expected that when applying more strict spatial constraints (30 km) that the agreement would be better, due to better co-location, which is not the case. Please explain.

Fig.1 lower panel: Moreover, why don't you make use of the maximum spatial resolution of IASI, i.e. comparing with the closest pixel or smaller radius especially as you mention city-scale studies earlier on.

Fig.1 lower panel: Please explain the spread of data points over the seasons. Why do you have significantly more points in spring than in summer and other seasons?

In this study, we found that the main source of atmospheric NH₃ in Paris is its transport from the surrounding agricultural regions resulting in higher correlations at larger scales between the ~12-km IASI footprints and the miniDOAS observations. Road-traffic emissions are noticeable only when agricultural related emissions are weak.

In addition, we have stated in the manuscript that the correlations at 30 km are not significant (p-value > 0.5) for two seasons (winter and fall) and the number of pairs are reduced by a factor of two compared to comparisons at 100 km.

IASI pixels size varies from 12-km diameter footprint on the ground at nadir to 40 km diameters depending on the satellite observational angle. For cloud free situations, IASI can measure the atmospheric radiation down to the ground so only the data that are not contaminated by clouds in the field-of-view were selected in this analysis. The variability of IASI pixels sizes and the presence of clouds can be seen for one day (3 May 2022) in Figure R2.

The trade-off between analysis box size and number of IASI samples justifies our choice of a 50 km window.

Finally, there are less pairs of comparison in fall and summer because the timeserie extends from January 2020 to May 31st 2022 with 166, 134, and 91 IASI days of observations in 2020, 2021, and 2022, respectively.

[Figure]

Figure R2: example of IASI morning observations with Metop B and C for one day (3 May 2022) in the domain of study.

We have added the following statements in the revised manuscript to discuss the spatial variability of $NH_3$ observed with the two datasets:

"The high correlation in spring between the two datasets can be attributed to two factors: 1) $NH_3$ concentrations are higher and therefore the signal measured by the two instruments are larger leading to a better correlation from the wide range of $NH_3$ concentrations (0-18 µg.m$^{-3}$ for the miniDOAS and 0-1.10$^{16}$ molecules.cm$^{-2}$ for IASI, Figure 1) and 2) the high amount of $NH_3$ emitted in spring in the surrounding regions due to fertilizer applications can be transported to Paris [Viatte et al., 2022; Viatte et al., 2021] resulting in high correlations between the ~12-km IASI footprints and the local miniDOAS observations."

"In fall and summer, the lower correlations between the ground-based and the satellite $NH_3$ observations could reveal specific $NH_3$ sources in the close vicinity of the miniDOAS which might be not representative of the IASI pixels size."

p.7 l.173: The satellite retrievals are strongly impacted by thermal contrast. It is highly surprising and difficult to believe that there is no impact of temperature on the correlation as it affects the sensitivity of the satellite data. On the other hand near-surface measurements are highly impacted by PBL height, while this is less the case for the column observations. Please explain.

We agree that figure S1 is not convincing and contradicts the statement that the miniDOAS observations are sensitive to the PBL height, while IASI are not, as they measure columns. Thus we have removed this statement and this figure in the revised supplementary information and manuscript documents.

p.7 l.177: If I understand it correctly, in the further analysis you average the IASI retrievals within a box of 50 km e.g. to compare with miniDOAS? In that case you cannot speak about a city-scale study neither about studying the spatiotemporal variability in Paris. Temporal yes, but not spatial!

Indeed, we have average IASI pixels within a 50km box around Paris taking into account the real elliptic size of each pixel as describe in Van Damme et al. (2018). The center of IASI pixels determines if it falls within the averaging box. We have modified "city-scale" to "Greater Paris-scale", "spatiotemporal" to "temporal"; and added this sentence to the revised manuscript: "we have selected coincident observations made within the same hour and the center of the IASI pixels was used to determine the distance between the miniDOAS and IASI measurements".

p.7 l.186: Please describe here or provide a reference how the error budget is done.

The uncertainties of the measurements are calculated following the literature recommendation, referenced in Volten et al., [2012]. We have added this reference in the manuscript "Description of the measurement uncertainties can be found in Volten et al. (2012)."

p.8 l.199: Agreed, but vertically you are also at large distance from the potential traffic emission source that you want to assess in this study, when not putting it at the surface. Official monitoring stations usually sample at 3 m altitude. This might also explain why you measure lower concentrations than in other urban areas / other studies.

We agree with the referee that at 40m in altitude, measurements are not affected by local traffic variability, and this could explain why we have lower concentrations that in other urban areas. The main goal of this study is to assess the $NH_3$ budget over the Greater Paris area, which is why we installed the instrument at the urban canopy so that the measurements footprint is representative of the Parisian urban area.

We have added a sentence in the revised manuscript to clarify: "The miniDOAS is located at an altitude of 40m so that its observation footprint is representative of the Greater Paris. This may partly explain the lower $NH_3$ concentrations observed in Paris compared to other urban areas."

p.15 l. 376: If the data from this traffic station (and maybe others) is available, why isn't it used in this study, even if it isn't a miniDOAS? It would make your results far more convincing, e.g. to check if the results are consistent for different stations and to assess the effect of having the miniDOAS at 40 m.

To our knowledge, there are no available $NH_3$ measurements in Paris close to traffic, and there are no other relatively long-term and high temporal resolution $NH_3$ observations in Paris at all. The reference we cited in this study (e. g., [Chatain et al., 2022]) is for another city of France (Reims). We clarified this in the revised manuscript. We would, indeed, by delighted to compare our $NH_3$ budget at 40m to the one close to the ground but unfortunately it is impossible for now.

Chatain, M., Chretien, E., Crunaire, S. and Jantzem, E.: Road Traffic and Its Influence on Urban Ammonia Concentrations (France), Atmosphere (Basel)., 13(7), doi:10.3390/atmos13071032, 2022.

Van Damme, M.; Clarisse, L.; Whitburn, S.; Hadji-Lazaro, J.; Hurtmans, D.; Clerbaux, C.; Coheur, P.-F. Industrial and agricultural ammonia point sources exposed. Nature 2018, 564, 99-103, doi:10.1038/s41586-018-0747-1.

Volten, H., Bergwerff, J. B., Haaima, M., Lolkema, D. E., Berkhout, A. J. C., van der Hoff, G. R., Potma, C. J. M., Wichink Kruit, R. J., van Pul, W. A. J. and Swart, D. P. J.: Two instruments based on differential optical absorption spectroscopy (DOAS) to measure accurate ammonia concentrations in the atmosphere, Atmos. Meas. Tech., 5(2), 413–427, doi:10.5194/amt-5-413-2012, 2012.

**Minor comments**

The minor revisions have all been considered in the revised manuscript.

p.4 l.88: Would be good to include as well https://amt.copernicus.org/articles/9/2721/2016/amt-9-2721-2016.pdf

p.8 l.202: I assume you provide first averages and after min max ranges? Please specify the numbers that are provided here.

We have specified that the first number is the average and the second one is the standard deviation.

p.8 l.212: Please elaborate shortly on these high pollution events.

We have modified the sentence: "IASI $NH_3$ total columns around Paris exhibit a higher $NH_3$ annual concentration and standard deviation in 2020 compared to the other years because of the multiple high pollution events occurring in spring during the 2020-COVID lockdown as described in Viatte et al. (2021)."

p.10 L.250: It is not really clear why this is then only the case in 2021, and for example not seen in 2020?

Interannual variability of $NH_3$ concentrations over Paris depends, in the first order, on agricultural activities and meteorological conditions. In September 2020, atmospheric temperatures were lower than in 2021 (see appendix S2) which could explain why the miniDOAS and IASI haven't monitor $NH_3$ enhancements. We have added this explanation to the revised manuscript: "In 2021, a second $NH_3$ enhancement is measured in September by the miniDOAS (2.73 ± 1.14 $\mu g.m^{-3}$) and IASI ($7.93 \times 10^{15}$ ± $4.64 \times 10^{15}$ $molecules.cm^{-2}$) which is not observed in 2020 possibly because atmospheric temperatures were lower than in 2021 (Figure S1)"

p.12 l.296: Please elaborate on why this is observed in agricultural areas. Is it related to reduced agricultural activity over the weekend?

Indeed, we have clarified this in the revised manuscript: "The weekly cycle of IASI measurements in Paris is almost analogous to the one observed over European agricultural areas with low concentrations observed on Mondays, as a result of reduced $NH_3$ emissions over the weekend, and an accumulation of ammonia during the week [Van Damme et al., 2022]."

**Technical corrections**

All technical corrections have been applied to the revised document.

p.1 l.20: in other urban areas

p.3 l.62: replace 'monitored' by 'detected'?

We replaced "monitored" by "observed".

p.7 l.156: do you mean quantitatively?

We have deleted this word in the revised manuscript.

p.11 l.278: 'raw' should be 'row'

p.15 l.380: I suggest replacing 'Atmospheric' by 'Temporal'